# Respiratory Dysfunction in Children and Adolescents with Mucopolysaccharidosis Types I, II, IVA, and VI

**DOI:** 10.3390/diagnostics10020063

**Published:** 2020-01-24

**Authors:** Assel Tulebayeva, Maira Sharipova, Riza Boranbayeva

**Affiliations:** 1Children’s diseases Department, S.D. Asfendiyarov Kazakh National Medical University, Almaty A05H2A6, Kazakhstan; 2General Pediatric Department, Scientific Center of Pediatrics and Pediatric Surgery, Almaty A15E2P4, Kazakhstan; mairash2004@mail.ru (M.S.); riza_brz@mail.ru (R.B.)

**Keywords:** mucopolysaccharidosis, respiratory dysfunction, pulmonary dysfunction, enzyme-replacement therapy, Kazakhstan

## Abstract

Mucopolysaccharidosis (MPS) is a rare genetic disease involving active storage of glycosaminoglycans (GAGs). Accumulation of GAGs in the connective tissues of airways leads to progressive pulmonary dysfunction. Studies conducted in Taiwan revealed mainly restrictive pulmonary dysfunction, whereas the same studies in Egypt and California revealed obstructive pulmonary dysfunction. The contradictory results and lack of studies of respiratory system in patients with MPS in Asian populations are an indication to study pulmonary impairment in patients with MPS in Kazakhstan. The prospective study of respiratory system in patients with MPS was conducted in the Scientific Centre of Paediatrics and Paediatric Surgery. Patients with MPS (*n* = 11) were examined for respiratory function. Different types of pulmonary dysfunction were present in MPS patients, they were mainly of a restrictive pathology. One patient with MPS II had obstructive dysfunction. Enzyme replacement therapy was provided for an average duration of four years, leading to improvements in respiratory function in two patients with total normalization in one. All observed patients had respiratory dysfunction, mainly of the restrictive type. Pulmonary impairment in patients with MPS is the main reason for death. Thus, it is necessary to follow up with pulmonary function assessments in children with MPS.

## 1. Introduction

Mucopolysaccharidosis (MPS) is a group of genetic disorders, caused by deficiency of lysosomal enzymes involved in the degradation of glycosaminoglycans (GAGs), thereby leading to progressive accumulation of GAGs [1,2] Seven types of MPS exist, and each is characterized by a specific lysosomal enzyme deficiency with the accumulation of different types of GAGs [3]

In patients with MPS, GAGs actively accumulate in connective tissues, leading to progressive multiorgan impairment [2]. Mortality in patients with MPS mainly results from progressive pathological processes in the cardiovascular and respiratory systems [4]. Involvement of the respiratory system in the pathological process of MPS has been shown to occur in 56–63% of cases [4,5]. Pulmonary dysfunction and airway obstruction in MPS are associated with morphological changes including shortness and joint instability of neck, high epiglottis, hypoplastic mandible, hypersalivation, mucoid hypersecretion, gingival hyperplasia, hypertrophy of the adenoids and tonsils, narrowed nasal airway, flattened nasal bridge, macroglossia, hepato-splenomegaly, tracheobronchomalacia and laryngomalacia, as well as skeletal deformations such as kyphoscoliosis, ribcage narrowing, temporomandibular joint ankylosis and chest wall deformity, decreased thoracic dimensions due to short stature, diaphragmatic weakness from spinal cord compression as a results of narrowed craniovertebral junction. The main cause that predisposes to airway obstruction is active GAG deposition in upper airway tissues, leading to distension of the tongue and hypertrophy of adenoids/tonsils, increased mucoid secretion also worsens this situation. Lower respiratory airway collapse usually appears due to laryngomalacia and tracheobronchomalacia as a result of GAG accumulation in the tracheobronchial cartilage. Respiratory impairment manifests clinically as frequently respiratory tract infections (sinusitis, rhinitis, otitis, bronchitis, and pneumonia) and obstruction of the upper airways. Upper airway dysfunction may also result from central nervous system pathology, including hydrocephalus and degeneration of neurons. In addition, active storage of GAGs leads to narrowing of the craniovertebral junction, resulting in spinal cord compression [2,6,7,8,9,10]. In contrast to obstructive conditions, restrictive respiratory failure is a consequence of the accumulation of GAGs in the alveoli and can also be a result of skeletal bone deformations, hepatosplenomegaly, weak respiratory muscles, stiffness of the ligaments, limited diaphragmatic excursion (due to enlarged liver), restriction of lung excursion, which typically occur in MPS I, II, IVA, and MPS VI [2,6,9,11]. According to Muhlebach, in Table 1 is presented respiratory manifestation in different types of MPS [6,7].

The clinical presentation of respiratory involvement in patients with MPS involves breathlessness, noisy breathing, coughing with difficult expectoration, wheezing, night snoring, frequent respiratory infections, and sleep apnoea. Obstructive sleep apnoea syndrome (OSAS) is estimated to occur in 69–95% of children with MPS [2,3] 

In a study conducted in Taiwan, involving 35 patients with MPS types I, II, III, IVA, and VI (aged 6.4–33 years), pulmonary pathology presented as restrictive disorders in the majority of individuals [12]. By contrast, in a study conducted in Egypt, involving 30 patients with MPS (aged 1–9 years), obstructive pulmonary dysfunction was the predominant type [13]. Similarly, in a study conducted in California, obstructive dysfunction was present in 15 out of 19 children with MPS [2]. However results of another study, revealed that among five MPS II patients (age range 29–50) three patients had mixed obstructive and restrictive airway disease [14]. These contradictory results, along with the limited number of observations in Asian populations, and the lack of data from such studies in Kazakhstan suggested that we should study pulmonary impairment in patients with MPS in Kazakhstan. 

Enzyme replacement therapy (ERT) is currently only available for MPS types I, II, IVA, VI, and VII. HSCT (haematopoietic stem cell transplantation) is a therapeutic option in patients with neurological involvement (under two years of life) because ERT does not cross the brain–blood barrier. Symptomatic therapy for respiratory system pathology in European countries consists of adenoidectomy, tonsillectomy, and continuous positive airway pressure (CPAP) [7,15]. In the Kazakhstan Republic, all children with diagnosed MPS types I, II, and VI are administered ERT. However, symptomatic therapy is provided only by the adenoidectomy and tonsillectomy. Treatment of OSAS with CPAP in children with MPS is not practised in Kazakhstan. 

Our aim was to identify the types of respiratory impairment and dysfunction that occur in Kazakhstani children with MPS types I (OMIM #607015), II (OMIM #309900), IVA (OMIM #253000), and type VI (OMIM #253200) and the effects of ERT on respiratory dysfunction.

## 2. Materials and Methods

### 2.1. Study Population

The study population consisted of 11 children and adolescents who were diagnosed with MPS types I, II, IVA, and VI, and who were under supervision at the Scientific Centre of Paediatrics and Paediatric Surgery. MPS diagnoses were made on the basis of clinical data, determination of lysosomal enzyme activities in leukocytes from dried blood spots, and genetic tests. None of the 11 study patients have cognitive impairment or behavioural problems. Patients with MPS II types have non-neuronopathic form of disease. One patient with MPS VII had a severely narrowed craniovertebral junction that led to wheelchair dependence. Five participants received ERT, with an average duration of four years (ranging from two to five years). Two patients with MPS I, two patients with MPS VI, and two siblings with MPS IVA had only recently been diagnosed and had not yet started ERT at the time of the study.

### 2.2. Ethical Approval 

All subjects gave their informed consent for inclusion before they participated in the study. The study was conducted in accordance with the Declaration of Helsinki, and the protocol was approved by the Ethics Committee of Kazakh National Medical University protocol no. 209 from 29.04.2015г. 

### 2.3. Enzyme Activity Levels

Determination of lysosomal enzymes activity levels in leukocytes was performed at the University Medical Centre Hamburg–Eppendorf, Germany. Lysosomal enzyme activity was measured in dried blood spots by fluorometry and tandem mass spectrometry in University Medical Centre (Hamburg–Eppendorf, Germany).

### 2.4. Genetic Analyses

Genetic analysis of genes *IDUA* (MPS I), *IDS* (MPS II), *GALNS* (MPS IVA), and *ARSB* (MPS VI) was carried out at Centogene (Rostock, Germany) (Table 2). Genes were analysed by polymerase chain reaction (PCR) and sequencing of the entire coding region and the highly conserved exon–intron splice junctions.

### 2.5. Pulmonary Function Testing

Evaluation of pulmonary function (spirometry) and cardiovascular function (echocardiography) was done at the time of diagnosis before ERT start for patients 1, 2, 6, 7, 10, and 11. While patients 3, 4, 5, 8, and 9 received ERT on average for 2.2 years. Pulmonary function was tested by spirometry with a BTL-08 Spiro (BTL, Newcastle-under-Lyme, UK), with participants in the standing position, with a closed nose, and rests between repeated tests. The following parameters were analysed: forced vital capacity (FVC), forced expired volume in 1 second (FEV1), the ratio of FEV1/FVC (Tiffeneau index), peak expiratory flow (PEF), and mid-expiratory flow (MEF) when 75% (MEF_75_), 50% (MEF_50_), and 25% (MEF_25_) of the FVC remains to be exhaled (Table 2). Normal values for these parameters are FVC > 80%, FEV1 > 75%, FEV1/FVC > 75%, MEF_75_, MEF_50_, and MEF_25_ > 75% of the reference value [16]. Obstructive dysfunction was defined as FEV1/FVC < 70% with FEV1 < 80%, and below normal values for MEF_75_, MEF_50_, and MEF_25_, indicating obstruction of large, medium, and small calibre bronchi, respectively. Restrictive dysfunction was characterized by low values for FVC and FEV1, and a normal or increased FEV1/FVC [16]. Physical exercise tolerance, as a marker of the functional status of the respiratory and cardiovascular systems, was assessed by a 6-min-walk test (6MWT).

### 2.6. Statistical Analyses

Descriptive statistics were calculated in Microsoft Excel 2007 for seven parameters: FVC, FEV1, the ratio of FEV1/FVC, MEF_75_, MEF_50_, and MEF_25_. 

## 3. Results

Of the 11 participants, eight were male. Nine were of Kazakh nationality, one was Russian (patient 3), and one was Turkish (patient 10). Four patients were diagnosed with MPS VI (Maroteaux–Lamy syndrome), three had MPS II (Hunter syndrome non-neuropathic), two had MPS I intermediate form (Hurler–Scheie syndrome) and two patients had MPS IVA (Morquio syndrome) (Table 3). At the time of examination, the mean age of the observed patients was 10.6 years (range 5–18 years), and the median age was 11 years. The mean age at the time of MPS diagnosis was 5.6 years (ranging from 1 year till 12 years). Due to the fact that the MPS is group of genetic diseases, the population included two siblings with MPS VI, two siblings with MPS IVA, and two cousins with MPS II. One child with MPS VI had consanguineous parents. Six participants had an adenoidectomy before the diagnosis of MPS. At the time of the study, six participants had adenoids, including one patient with MPS VI, who previously underwent adenoidectomy and had the recurrence of adenoids. All individuals had short stature and frequent respiratory infections, such as rhinitis, bronchitis, and pneumonia. 

All 11 individuals had chest deformations, ranging from mild to severe kyphosis of the thoraco-lumbar part of the spine (*n* = 11). Nine individuals had hypersalivation and gingival hyperplasia, and eight had macroglossia. Lung auscultation revealed harsh vesicular breathing in nine individuals, and wheezes and crepitations in four. Eight of the individuals complained of noisy nasal breathing, shortness of breathing, and snoring during sleep. Clinically, there were no signs of active respiratory infection. Hepatomegaly was revealed in six individuals by abdominal ultrasonography.

Cardiovascular examination with echocardiography revealed left-ventricular hypertrophy in six patients, mitral- and aortic-valve impairment (with I–III degree regurgitation) in all patients.

The mean walking distance among 10 patients who passed the test for physical exercise tolerance was 282.3 m (ranging from 100 to 400 m). Patient 8 with MPS VI could not pass the test, his distance was only 5 m. 

Mixed pulmonary dysfunction with predominantly restrictive pathology was identified in 10 patients. Mean forced vital capacity (FVC) and forced expired volume in 1 second (FEV1) in these patients were 50.3% and 54.9%, respectively, compared with normal values of > 80% and > 75%, respectively. Obstructive type of dysfunction was observed only in patient 3 with MPS II (Table 4).

Spirometry results during an average 1.8 years of follow-up were obtained for five individuals: three with MPS II (patients 3, 4, 5) and two siblings with MPS VI (patients 8 and 9, Table 1). Among these individuals, spirometry results worsened in two patients. Patient 3 with MPS II (Table 1), who was diagnosed with obstructive disorder, showed reductions in all parameters: FVC by 54.4%, FEV1 by 53.2%, MEF_75_ by 51.3%, MEF_50_ by 56.9%, and MEF_25_ by 46.4% (Table 3). The second patient (patient 8) with worsening spirometry results had MPS VI and was diagnosed with predominantly restrictive disorder (Table 3). His spirometry parameters had a negative trend, with reduction of FVC by 51.2%, FEV1 by 51.2%, MEF_75_ by 29.1%, and MEF_25_ by 56.4%. The FEV1/FVC remained constant in this patient with slight increase of 5.2% in MEF_50_. By contrast, two patients (one each with MPS II and MPS VI) had improvements in respiratory function, with an average rising of FVC increase of 18%, mean FEV1 increase of 18.7%, and mean growth in MEF_75_, MEF_50_, and MEF_25_ by 48.7%, 16.1%, and 21%, respectively (Table 5). One MPS II patient’s parameters were totally normalized (Table 5). 

In the 6-min-walk test (6MWT) on ERT, patients 3, 4, 5, and 9 had a positive trend, and the distance walked increased on average by 141 m. The exception was patient 8 with severe, progressive MPS VI, who earlier was able to walk only 5 m, after 1 year of follow up he completely lost the ability to walk. This patient also had a worsening of pulmonary function. The negative trend may have been caused by progressive craniovertebral junction narrowing and resulting spinal cord compression, which was confirmed by magnetic resonance imaging. 

## 4. Discussion

MPS consists of a group of lysosomal storage diseases that are characterized by active accumulation of GAGs in all tissues, mainly in connective tissue structures. Respiratory complications are observed in all types of MPS and are a common cause of death in children with this disease. Limited data exist regarding respiratory system impairment in Asian children with MPS. Our study has now provided the first observational results from the Kazakhstan Republic of pulmonary pathology in children with MPS. 

Results from research conducted in California have demonstrated improvements in respiratory function in children with MPS VI after 72–96 weeks of ERT, with 17% and 11% increases in FVC and FEV1, respectively, compared with initial measurements [17]. Results of phase III clinical trials of ERT for MPS I showed clinical improvements at six months of 5.6% compared with the placebo group, and after 3.5 years of receiving ERT, the FVC (%) rate remained the same [17].

Impairment of the pulmonary system in patients with MPS is associated with many factors, including the typical morphophenotypes of hypertrophy of the adenoids and tonsils, laryngomalacia, tracheobronchomalacia, obstructive and restrictive lung disease, small chest volume, skeletal deformation, and hepatomegaly. It should be noted that these factors worsen with age and cause progression of respiratory disease. 

In a study of five adult patients with MPS II, examination of computed tomography (CT) scans identified median levels of collapse of the trachea of 68% 1 cm above the aortic arch, 64% at the level of the aortic arch, and 58% 2 cm above the trachea bifurcation [11]. Bronchial collapse below the trachea bifurcation occurred at a level of 58% in the left main bronchus and 44% in the right bronchus. Respiratory function tests by spirometry indicated that all patients had an obstructive type of respiratory dysfunction, probably resulting from tracheal collapse. In comparison, the results of a study of younger patients with MPS II (with a mean age of 9 years) identified predominantly restrictive respiratory failure [13]. In a study of 30 patients (aged 1–9 years) with MPS types I, II, III, and IV, obstructive pulmonary impairment was observed in 66.6% of patients, whereas 20% had a combined obstructive and restrictive pathology [13]. In a study of 35 patients (mean age 14.6 years) with MPS types I, II, IVA, and VI, restrictive pulmonary pathology was observed in 48% of patients, whereas obstructive dysfunction was observed in only 9% (two adults aged 24 and 33 years) [14]. Follow-up spirometry was conducted on eight of the patients who had mild forms of either MPS II or MPS VI and who had received ERT for 1.5–7.4 years. These eight patients had increases in the FVC and FEV1 parameters from pre-treatment levels, with trends towards normalization. In these 35 patients, spirometry parameters (FVC, FEV1, peak expiratory flow (PEF), and FEV1/FVC) were negatively correlated with age (*p* < 0.01), so pubertal and post-pubertal patients had more severe impairment of the respiratory system than pre-pubertal patients [14]. The negative correlation between age and the function of the respiratory system could be due to a worsening of obstruction, because of progressive tracheobronchomalacia. Our results have also demonstrated a predominance of restrictive pathology of the respiratory system in young patients with MPS. Only one patient (patient 3) in our population (an 18 year old with MPS II) had obstructive pathology, and the spirometry values worsened with age in this patient, in spite of the use of ERT. However, a CT scan was not performed in this patient to clarify the presence of tracheobronchomalacia, which can lead to airway obstruction. A notable limitation of the use of spirometry to study the respiratory system is that children with severe neuropathic forms of MPS are not able to perform all the required tasks. According to the results of research work, done by Paul Harmatz et al., ERT improves respiratory function directly and also through the increasing growth, which changes lung function (FVC) [17]. In our study, we mentioned only baseline growth and did not do the correlation between growth change and respiratory function improvement during ERT. 

## 5. Conclusions

All patients in our study population had respiratory dysfunction. The majority had mixed respiratory pathology, predominantly of the restrictive type. In summary, our results suggest that ERT has mixed effectiveness in paediatric patients with MPS in a Kazakhstani population. A patient with MPS II and a patient with MPS VI had progression of respiratory pathology. However, two patients with MPS II and one patient with MPS VI demonstrated some improvement on ERT. The average duration of ERT of the mentioned patients was 3.5 years. The mean duration of treatment of all patients before respiratory evaluation was approximately 2.2 years. Because pulmonary pathology carries a high risk of mortality in patients with MPS and this pathology worsens with age, children and adolescents with MPS should be followed up with spirometry and CT scans of the respiratory system. 

## Figures and Tables

**Table 1 diagnostics-10-00063-t001:** Respiratory manifestation in different types of mucopolysaccharidosis (MPS).

MPS Type	Upper Airway Obstruction	Lower Airway Obstruction	Restrictive Lung Disease
I	+++	+++	+++
II	+++	+++	++
III	+	+	+
IVA	++	+	+++
VI	+++	+++	++
VII	+++	+++	++

“+” minimal, “++” moderate, “+++” severe respiratory manifestation.

**Table 2 diagnostics-10-00063-t002:** Genetic characteristics of the study group.

Patient	MPS Type	Gene	Mutation
1	I	*IDUA*	c.208C>T
2	I	*IDUA*	c.1598C>T c.1709A>T
3	II	*IDS*	c.776T>G
4	II	*IDS*	c.1000G>A
5	II	*IDS*	c.1000G>A
6	IVА	*GALNS*	c.463G>A c.1462G>A c.572A>G
7	IVА	*GALNS*	c.463G>A c.1462G>A c.572A>G
8	VI	*ARSB*	c.275C>A
9	VI	*ARSB*	c.275C>A
10	VI	*ARSB*	c.1544C>T
11	VI	*ARSB*	c.275C>A

**Table 3 diagnostics-10-00063-t003:** Clinical characteristics of the study group.

Patient	Age (Years)	Sex	MPS Type	Age at Diagnosis (Years)	Height (cm)	Weight (kg)	ERT	Age at ERT Start (Years)
1	11	f	I	1	110	18	no	
2	5	f	I	5	105	16	no	
3	18	m	II	2	140	44	yes	17
4	7	m	II	6	105	20	yes	6
5	5	m	II	4	113	21	yes	4
6	10	m	IVА	8	116	26	no	
7	12	m	IVА	10	117	22	no	
8	14	m	VI	3	93	18	yes	10
9	15	m	VI	4	120	34	yes	11
10	8	f	VI	7	86	12	no	
11	12	f	VI	12	97	19	no	

ERT—enzyme replacement therapy.

**Table 4 diagnostics-10-00063-t004:** Spirometry results (%) in children and adolescents with different types of MPS.

Patient	MPS Type	Age (Years)	Height (cm)	FVC	FEV1	FEV1/FVC *	MEF75	MEF50	MEF25	PEF
1	I	11	110	46.92	52.58	117.87	78.61	85.12	86.49	78.2
2	I	5	105	62.37	68.47	118.39	71.25	76.87	77.64	78.8
3	II	18	140	93.65	77.31	85.48	43.25	45.74	71.97	35.2
4	II	7	105	37.3	38.92	109.77	45.14	39.43	36.49	54.8
5	II	5	113	40.49	45.58	118.39	46.41	52.28	58.91	47.5
6	IVА	10	116	66	74.3	118.39	149.59	153.21	146.96	165
7	IVА	12	117	70.4	74.1	110.01	87.7	68.2	71.69	100
8	VI	14	93	36.2	40.77	118.39	44.89	39.61	86.32	48.4
9	VI	15	120	59.18	66.72	118.39	50.6	97.97	88.59	77.5
10	VI	8	86	53.91	57.87	115.76	69.65	56.02	51.82	68.3
11	VI	12	97	27.34	30.01	118,39	30.52	36.27	32.95	33.7
Mean		10.6		54.7	56.9	113.9	65.2	68.2	73.6	71.6
SD		4.15		19.9	16.4	10.35	33	34.7	31.1	37.19

Normal > 80% of the reference value; * normal > 75% of the reference value; FVC—forced vital capacity, FEV1—forced expired volume in 1 second, the ratio of FEV1/FVC (Tiffeneau index), PEF—peak expiratory flow, MEF—mid-expiratory flow when 75% (MEF_75_), 50% (MEF_50_), and 25% (MEF_25_) of the FVC remains to be exhaled.

**Table 5 diagnostics-10-00063-t005:** Spirometry results (%) of patients with MPS II and VI on ERT.

Patient	MPS Type	ERT Duration at the Beginning of Study (Weeks)	Total ERT Duration (Weeks)	FVC (%)		FEV1 (%)		MEF_75_ (%)		MEF_50_ (%)		MEF_25_ (%)	
				1	2	1	2	1	2	1	2	1	2
Patient 3	II	52	208	93.6	42.6	77.3	36.1	43.2	21	45.7	19.6	71.9	38.5
Patient 4	II	52	208	37.3	67.8	38.9	70	45	101	39.4	69.4	36.4	55.9
Patient 5	II	52	104	40.5	81.3	45.5	84.5	46.4	95	52.3	95	58.8	75.6
Patient 8	VI	208	260	59.2	64.8	66.7	73.7	50.6	91.2	97.9	99.7	88.6	111
Patient 9	VI	208	260	36.2	17.6	40.7	19.8	44.9	31.8	39.6	41.7	86.3	37.6

1—dates at the beginning of study; 2—dates after follow-up.

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
