# Peer review of "Respiratory Dysfunction in Children and Adolescents with Mucopolysaccharidosis Types I, II, IVA, and VI"

_diagnostics, 2020, doi:10.3390/diagnostics10020063_

Round 1
Reviewer 1 Report
The authors present a study on the respiratory function of the MPS population in Kazakhstan. To the best of my knowledge this is the first study on this subject in Asian MPS patients. The overall manuscript is easy to read, with an up-to-date introduction and well-focused discussion, comparing the present results with others of the same kind performed by other teams worldwide.
Overall, I have no doubts in recommending it for publication. There are only some few minor points that the authors should try to correct and/or address. I have also a few suggestions that the authors may want to consider:
* I would like to start by a general comment on the way the authors present the MPS diseases throughout the manuscript. On most sections the authors (except the discussion) say that MPS is a rare genetic disease. I tend to consider this description as inaccurate, as it is actually a GROUP of genetic diseases, as the authors themselves eventually point out. So, for the sake of accuracy, I would change a few sentences by referring this fact from the beginning: we’re not talking about a single clinical entity, even though there are many common points between the different MPS. I would recommend the authors to review the following sentences:
Abstract, page 1, line 9; Introduction, page 1, line 27; Results, page 2, line 76
* In order to the patients’ characterization to be more complete, on table 1, I would add a column with the sex of each patient and disclose which ones were of non-Kazakh nationality. This last information may be given directly in the text (p.e., on the results section, page 2, lines 72-73, just add the number of each patient between parenthesis after Russian and Turkish).
* I know all the clinical acronyms have been disclosed on the materials and methods section at the end of the manuscript but I felt the need to be constantly checking them while reading tables 2 and 3 and the results section - and it didn’t fell practical at all. So, I would recommend the authors to disclose the acronyms already on page 3 and 4 (as a footnote below the tables, for example) and then use only the acronyms alone latter, on the M&M section.
* Finally, I think the paper could also benefit from an extra table with info on the molecular basis of disease in the studies cases. I know the major issue underlying this study is the overall understanding of the pulmonary dysfunction on MPS patients and their genetic characterization lies beyond this scope. Still, as a molecular biologist, I would like to see the genotypes of these patients referred somewhere along the manuscript. Considering the M&M section does include a paragraph on genetic analysis (4.2), my personal suggestion to the authors would be to include a supplementary table with the genetic characterization of the 11 patients. I think it would give the readers the possibility to check for genotype-phenotype correlations, or even for clinicians who have patients with the same genotype to compare the clinical observations here described with their own. Overall, I fell that in this sort of rare genetic syndromes, the more info we share on the patients and their clinical course, the more we can learn… Still, this is my personal opinion, and I do not fell one such table is mandatory for the manuscript to be accepted.
Other than that, I have only found one minor point that should be corrected:
On page 5, line 174 (M&M section) the authors should use the MPS acronym instead of “mucopolysaccharidosis”.
Author Response
Response to Reviewer 1 Comments
Dear Reviewer many thanks for analysis of my manuscript and for recommendations!
I did corrections in manuscripts according to Your recommendations.
* I would like to start by a general comment on the way the authors present the MPS diseases throughout the manuscript. On most sections the authors (except the discussion) say that MPS is a rare genetic disease. I tend to consider this description as inaccurate, as it is actually a GROUP of genetic diseases, as the authors themselves eventually point out. So, for the sake of accuracy, I would change a few sentences by referring this fact from the beginning: we’re not talking about a single clinical entity, even though there are many common points between the different MPS. I would recommend the authors to review the following sentences:
Abstract, page 1, line 9; Introduction, page 1, line 27; Results, page 2, line 76
Response 1: the “GROUP of genetic diseases” part was added in abstract, page 1, line 9; Introduction, page 1, line 2; Results, page 2, line 55.
* In order to the patients’ characterization to be more complete, on table 1, I would add a column with the sex of each patient and disclose which ones were of non-Kazakh nationality. This last information may be given directly in the text (p.e., on the results section, page 2, lines 72-73, just add the number of each patient between parenthesis after Russian and Turkish).
Response 2: On the table 1 I added the column with the sex of each patient and results section, page 2, line 50 the number of each patient of Russian and Turkish nationality was added.
* I know all the clinical acronyms have been disclosed on the materials and methods section at the end of the manuscript but I felt the need to be constantly checking them while reading tables 2 and 3 and the results section - and it didn’t fell practical at all. So, I would recommend the authors to disclose the acronyms already on page 3 and 4 (as a footnote below the tables, for example) and then use only the acronyms alone latter, on the M&M section.
Response 3: I added acronyms of spirometry on the footnote below the table 2.
* Finally, I think the paper could also benefit from an extra table with info on the molecular basis of disease in the studies cases. I know the major issue underlying this study is the overall understanding of the pulmonary dysfunction on MPS patients and their genetic characterization lies beyond this scope. Still, as a molecular biologist, I would like to see the genotypes of these patients referred somewhere along the manuscript. Considering the M&M section does include a paragraph on genetic analysis (4.2), my personal suggestion to the authors would be to include a supplementary table with the genetic characterization of the 11 patients. I think it would give the readers the possibility to check for genotype-phenotype correlations, or even for clinicians who have patients with the same genotype to compare the clinical observations here described with their own. Overall, I fell that in this sort of rare genetic syndromes, the more info we share on the patients and their clinical course, the more we can learn… Still, this is my personal opinion, and I do not fell one such table is mandatory for the manuscript to be accepted.
Response 4: I added table â„– 4 with genetic characterization (mutations) of each patient.
On page 5, line 174 (M&M section) the authors should use the MPS acronym instead of “mucopolysaccharidosis”.
Response 5: I put on page 5, line 190 MPS acronym instead of “mucopolysaccharidosis”.

Reviewer 2 Report
Congratulation to the authors for the aim to report their experinece about MPS. This is a very pressing theme: respiratory problems involve dramatically all patients especially who have no neurological problems.
Moderate English language changes are required.
I suggest to review the abstract:
the period of follow-up is not well specified in the abstract and also in the discussion. I suggest to specify which type of MPS are affected the patients Avoid repeating phrases and I appreciate if you would clarify and simplify the textAt line 27 (introduction) the correct form is "caused by". The type of inheritance wold be mention and also the fact that GAGs are ubiquitous.
At line 30: Gags and not GAGSIO (error typing).
At line 36 I suggest to add macroglossiaa and hepato-splenomegaly; I suggest also to change "joint instability of neck" rather "than immobile neck".
At line 62: Also MPS VII is now treatable with ERT. I suggest to add that HSCT (Haematopoietic stem cell transaplantation) is a therapeutic options in patients with neurological involvement (under 2 years of life) because ERT do not cross brain blood barrier.
In the results the age at diagnosis is not report: it is available in the table 1 but not in the text. You may explain better at line 74 that Hurler-Scheie is the intermediate forma and also you can specify the severity of MPS II patients.
The motivation about some patients are not treated with ERT is not well understandable.
The results from line 100 to line 112 should be explain better.
At line 122 "mainly" instead of "primarily".
The discussion is well conducted but the results of the present study are mixed up to the results of other papers: you may distinguish and elaborate better the last part from line 156.
In the conclusion the duration of ERT (weeks of ERT) before respiratory evaluation (at line 207) is missed.
The limits of this study are:
short observation period eterogeneous population in trem of type of disease, severity and therapy (often ERT was started late with severe tissue involvement in a progressive disease as MPS)You could include in the references the following paper:
Bianchi PM, Gaini R, Vitale S. ENT and Mucopolysaccharidoses, Italian Journal of Paediatrics 2018;44:127
Author Response
Response to Reviewer 2 Comments
Dear Reviewer many thanks for analysis of my manuscript and for recommendations!
I did corrections in manuscripts according to Your recommendations.
*the period of follow-up is not well specified in the abstract and also in the discussion. I suggest to specify which type of MPS are affected the patients Avoid repeating phrases and I appreciate if you would clarify and simplify the text
Response 1: I added in abstract and discussion the period of follow-up.
*At line 27 (introduction) the correct form is "caused by".
Response 2: At line 27 (introduction) the correct form is "caused by" was written.
*At line 30: Gags and not GAGSIO (error typing).
Response 3: At line 18 I corrected typing “GAGs”.
*At line 36 I suggest to add macroglossiaa and hepato-splenomegaly; I suggest also to change "joint instability of neck" rather "than immobile neck".
Response 4: I added at line 13 macroglossia and hepato-splenomegaly; and changed "joint instability of neck" rather "than immobile neck".
*At line 62: Also MPS VII is now treatable with ERT. I suggest to add that HSCT (Haematopoietic stem cell transaplantation) is a therapeutic options in patients with neurological involvement (under 2 years of life) because ERT do not cross brain blood barrier.
Response 5: At line 37-38 I added that MPS VII is also now treatable with ERT and added following sentence: that HSCT (Haematopoietic stem cell transplantation) is a therapeutic options in patients with neurological involvement (under 2 years of life) because ERT do not cross brain blood barrier.
*In the results the age at diagnosis is not report: it is available in the table 1 but not in the text. You may explain better at line 74 that Hurler-Scheie is the intermediate forma and also you can specify the severity of MPS II patients.
Response 6: In the results section I added the age at diagnosis in the text; also I specified at line 74 that Hurler-Scheie is the intermediate form and specified the severity of MPS II patients.
*The results from line 100 to line 112 should be explain better.
Response 7: A bit changed the text of results from line 78 to line 89.
*At line 122 "mainly" instead of "primarily".
Response 8: At line 99 wrote "mainly" instead of "primarily".
*In the conclusion the duration of ERT (weeks of ERT) before respiratory evaluation (at line 207) is missed.
Response 10: In the conclusion the duration of ERT (weeks of ERT) before respiratory evaluation (at line 187) was added.
*You could include in the references the following paper: Bianchi PM, Gaini R, Vitale S. ENT and Mucopolysaccharidoses, Italian Journal of Paediatrics 2018;44:127.
Response 11: Also added in the references: Bianchi PM, Gaini R, Vitale S. ENT and Mucopolysaccharidoses, Italian Journal of Paediatrics 2018;44:127 under â„–10 in the list.